# The Effect of Different Glucose Concentrations on the Antiproliferative Activity of Metformin in MCF-7 Breast Cancer Cells

**DOI:** 10.3390/pharmaceutics15092186

**Published:** 2023-08-23

**Authors:** Sholpan Nurzhan, Zhibek Bekezhankyzy, Hong Ding, Nurken Berdigaliyev, Shynggys Sergazy, Alexander Gulyayev, Zarina Shulgau, Christopher R. Triggle, Mohamad Aljofan

**Affiliations:** 1Department of Biomedical Sciences, School of Medicine, Nazarbayev University, Astana Z05H0P9, Kazakhstan; sholpan.nurzhan@nu.edu.kz (S.N.); zhibek.bekezhankyzy@nu.edu.kz (Z.B.); nurken.berdigaliyev@nu.edu.kz (N.B.); 2National Center for Biotechnology, Astana Z05K8D5, Kazakhstan; shynggys.sergazy@nu.edu.kz (S.S.); akin@mail.ru (A.G.); 3Department of Pharmacology, Weill Cornell Medicine in Qatar, Education City, Doha P.O. Box 24144, Qatar; hod2005@qatar-med.cornell.edu (H.D.); cht2011@qatar-med.cornell.edu (C.R.T.); 4Drug Discovery and Development Laboratory, National Laboratory Astana, Nazarbayev University, Astana Z05H0P9, Kazakhstan; 5Research Institute of Balneology and Medical Rehabilitation, Akmola Region, Burabay 021708, Kazakhstan

**Keywords:** metformin, apoptosis, antiproliferative, mTOR, BCL-2, glucose

## Abstract

The glucose-lowering drug metformin has been reported to have anticancer properties through unknown mechanisms. Other unknown factors that may influence its anticancer potential include the glycemic status of the patient. Therefore, the objective of this study is to determine the effect of different glucose environments on the antiproliferative potency and the cellular mechanism of action of metformin. Human breast cancer cells, MCF-7, were incubated in low, normal, elevated, and high glucose environments and treated with metformin. The antiproliferative potential of metformin and its effect on protein expression as well as its ability to induce cellular apoptosis and autophagy under different glucose environments, were determined using different molecular techniques. Metformin significantly inhibited cellular proliferation in a time- and glucose-concentration-dependent manner. In comparison to elevated glucose, low normal glucose alone induced a significant level of autophagy that was further increased in the presence of metformin. While glucose concentration did not appear to have an effect on the antiproliferative potency of metformin, the cellular basis of action was shown to be glucose-dependent. The antiproliferative mechanism of action of metformin in elevated and low normal glucose environments is mTOR-dependent, whereas, in the high glucose environment, the antiproliferative mechanism is independent of mTOR. This is the first study to report that both the antiproliferative potency and the cellular mechanism of action aredependent on the concentration of glucose.

## 1. Introduction

In 2005, a retrospective study from Tayside, Scotland, concluded that patients with type 2 diabetes treated with metformin had a lower risk of cancer. Subsequently, similar conclusions have been reached by many other investigators. For instance, see Bodmeretal., 2010 [1] and also reviewed by Triggle et al., 2022 [2].

The cellular mechanism whereby metformin exerts its antiproliferative actions remains unclear, with some reports linking activity to its glucose-lowering effect, inferring a benefit in slowing tumor proliferation in people with hyperglycemia [3]. Other reports have argued that metformin targets the respiratory complex I of the electron transport chain in the mitochondria of preneoplastic and neoplastic cells and thereby reduces cellular energy consumption [4]. There are two important signaling pathways; the insulin/insulin-like growth factor-1 (IGF1) signaling pathway is activated when nutrients are available, and the adenosine mono-phosphate protein kinase (AMPK) pathway is activated during a lack of nutrients. Both pathways are important negative regulators of the mammalian target for the rapamycin (mTOR) pathway, which represent potential key targets for mediating the antiproliferative effects of metformin [5,6,7].

Impaired glucose metabolism is a risk factor for the development of cancer in diabetic patients [8]. A study by Wahdan-Alaswad et al. suggested that augmented glucose concentrations promote breast cancer proliferation and reduce the anticancer effectiveness of metformin [9]. Furthermore, studies have shown that lowering the glucose accessible to breast cancer cells improves the chemotherapeutic response to metformin [10] and that metformin can be lethal to breast cancer cells and cancer stem cells under glucose-starved conditions [11]. This might be due to the fact that cancer cells have higher metabolic demands than normal cells, and hence, starving them by cutting off their energy sources and targeting metabolic reprogramming in cancer cells can specifically decrease cancer cell proliferation and thereby suppress tumor growth [12].

In order to maintain their uncontrolled growth rate, cancer cells have developed a novel method of obtaining energy from the outside. For example, in the presence of oxygen, normal cells use glycolysis to metabolize glucose, but during glucose restriction, they convert glucose to lactic acid. However, cancer cells convert glucose to lactate in the presence of oxygen (Warburg effect), and most cancer cells rely on glycolysis to create a significantly lower amount of ATP [13], which is one of the first biochemical hallmarks of cancer cells.

In support of this theory, Zordoky et al. reported that metformin in the concentration range of 500 μM to 5 mM under normal glucose conditions (5 mM) resulted in significant suppression of cell growth, whereas raising glucose (25 mM) reduced the antiproliferative effects of metformin in the triple-negative MDA-MB-231 breast cancer cell line [14]. In contrast, Liu and colleagues who used renal carcinoma cell lines (A498 and GRC-1) indicated that under normal glucose conditions, metformin was able to inhibit cellular proliferation, but with glucose starvation, treatment with 3 mM metformin enhanced proliferation via an AMPK-dependent mechanism [5]. Interestingly, Tossetta et al., reported that metformin could increase cancer cell sensitivity to chemotherapeutics in a glucose-independent manner [15].

In conclusion, it is not clear which glucose environment will enhance the antiproliferative activity of metformin. Determining the most suitable glucose environment for the antiproliferative activity of metformin would improve its potential clinical use as an anticancer drug. Thus, the aim of this study is to examine the effect of different glucose environments on the antiproliferative activity of metformin in human breast cancer cells (MCF-7).

## 2. Materials and Methods

### 2.1. Cellular Proliferation Assay

Cellular proliferation assay using MCF-7 cells was determined using the 3-[4,5-dimethylthiazol-2-yl]-2,5-diphenyltetrazoliumbromide (MTT) method according to manufacturing guidelines. Briefly, cells were seeded at 1 × 10^4^ cells/100 µL in 96-well microtiter plates in Dulbecco’s Modified Eagle’s Medium (Sigma-Aldrich, St. Louis, MO, USA) with essential nutrients andincubated overnight for attachment. After 20 h, a final volume of 100 μL of serial log dilutions of metformin and 5% DMSO (positive control) diluted in media with different glucose concentrations (available from Sigma-Aldrich); low normal glucose (NG 1 g/L glucose), elevated glucose (EG 2 g/L glucose), and high glucose (HG 4.5 g/L glucose), added to each well in triplicate (final compound concentrations ranged from 9 μg/mL to 150 μg/mL, equivalent to human dose range of approximately 300 mg–5 g/day [16]). These concentrations of glucose were chosen as representing those seen for patients without diabetes and representing normoglycemia (low normal, NG, at 5.5 mM), elevated (EG, 11 mM) as might be observed in patients with diabetes, and high glucose (HG at 25 mM) as in uncontrolled hyperglycemia. Comparable concentrations of glucose have been used in other studies (see Zordoky et al., 2014) [14]. MCF-7 cells were incubated in the presence of metformin and DMSO at 5% CO_2_ at 37 °C for 24 h and 48 h. MCF-7 cells were then treated with MTT (Sigma-Aldrich, St. Louis, MO, USA) and incubated for 4 h. After the incubation period, formazan crystals were dissolved by DMSO, and the absorbance was measured at 570 nm using a 96-well imaging reader (iMark Microplate reader, Bio-Rad, Inc., Hercules, CA, USA). The cytotoxicity was determined using the untreated cells as the negative control. For a comparison between normal and malignant cells, please refer to Bekezhankyzy et al., 2023 [17]. The percentage of antiproliferative activity (in treated cells) and cytotoxicity (in non-treated cells) were calculated according to the method described by Florento et al., 2012 using the treated reading/negative control (non-treated); cell viability (%) = [(A_sample at 570 nm_/A_control at 570 nm_)] × 100 [18].

### 2.2. Oxidative Stress Assay

MCF-7 were treated with metformin (85 µM) in NG and EG glucose environment and incubated for 24 h at 5% CO_2_ at 37 °C. The oxidative status was determined using FRAS 4 device (Evolvo S.R.L., Cinisello Balsamo, Italy) and d-ROMs Test kits, which show the levels of free radicals. The antioxidant was measured using the Plasma Antioxidant Test kit (PAT test) (H&D S.R.L., Parma, Italy, RDP.50.MG1903), which is a photometric test that enables the determination of the total antioxidant activity of blood plasma. 

### 2.3. Western Blotting

MCF-7 cells were incubated for 20 h and then treated with metformin (85 µM) in different glucose media (NG, EG, and HG). After 24 and 48 h of treatment, the cells were harvested with 100–150 µL of Laemli Buffer and lysed using loading dye containing 10% glycerol and β-mercaptoethanol and bromophenol blue in a 1:25 ratio. Protein concentrations were determined by BCA assay (71285-3, Novagen), and equal quantities of protein (10 μg) were loaded on two different gels, 12% (for small proteins) and 6% (for >200 kDa proteins) SDS-PAGE, and transferred topolyvinylidene fluoride (PVDF) membranes. The membranes were incubated in a blocking buffer (Tris-buffered saline [TBS] buffer containing 0.1% Tween 20 [Sigma] and 5% nonfat dry milk) for 1 h at room temperature to block nonspecific binding. The membranes were then incubated with primary antibodies: anti-alpha-tubulin (T6199, Sigma-Aldrich, diluted 1:10,000), anti-BCL-2 (SAB4300339, Sigma Aldrich, diluted 1:800) and anti-mTOR (T2949 by Sigma-Aldrich, diluted 1:2000) at 4 °C overnight. The membranes were washed in TBST and incubated with horseradish peroxidase (HRP)-conjugated secondary antibodies anti-rabbit (A0545, Sigma-Aldrich, diluted 1:5000) and anti-mouse (A9044, Sigma-Aldrich) for 1 h at room temperatureand then detected using anenhanced chemiluminescence (ECL) detection system. For band intensity comparisons and quantification, the Image Analyzer (BIO-RAD) was used.

### 2.4. Autophagy Assay

Cellular autophagy measured using an Autophagy Assay kit (Abcam, Cambridge, UK) according to the manufacturer’s guidelines. Cells were cultured in a black 96-well plate with a clear bottom at a density of 1.5 × 10^4^ cells/well, then treated with the metformin (concentration of 85 µM) in three different media (NG, EG, and HG) and incubated at 5% CO_2_ at 37 °C for 24 h. Following the incubation period, the medium was removed from the cells, and100 μL of the autophagosome detection reagent working solution wasadded to each well (samples and controls) and then incubated at 37 °C with 5% CO_2_ for 40 min. Cells were washed four times by gently adding 100 μL of wash buffer to each well and were then carefully removed to prevent dislodging the cells. Fluorescence intensity was then measured using an imaging reader (Cytation^TM5^, Bio-Tek Instruments, Agilent Technologies, Inc., Santa-Clara, CA, USA) at λ_ex360_/λ_em520_ nm. Cell samples in the wells were captured with DAPI.

### 2.5. Cellular Apoptosis

Apoptosis of the breast cancer cells following metformin treatment was determined using a Caspase 3/7 apoptosis detection kit (Catalogue No. MCH100108, EMD Millipore Corporation, Billerica, MA, USA) according to the manufacturer’s protocol. Briefly, MCF-7 cells (0.8 × 10^6^) were plated in each well of a 6-well plate and allowed to reach the exponential growth phase before being treated with metformin (85 µM), DMSO, and non-treated control for 24 h and 48 h. The cells were then harvested and collected by centrifugation at 400× *g* for 5 min at 4 °C, and resuspended in 400 µL Assay buffer BA. Next, 50 µL of cell suspensions were incubated with Caspase 3/7 Working Reagent for 30 min in the incubator at 37 °C with 5% CO_2_; following incubation, 150 µL of the Muse Caspase 7-AAD working solution was added and placed at room temperature for 5 min in the dark. Apoptosis was measured by Muse Cellular Analysis machine (Guava Muse Cellular Analyser, LO86, Luminex, Austin, TX, USA).

### 2.6. Mitopotential Assay

MCF-7 breast cancer cells (0.8 × 10^6^) were plated in a 6-well plate and allowed to reach the exponential growth phase and then treated with metformin (85 µM), DMSO, and non-treated control for 24 h and 48 h. Cells were then harvested and collected by centrifugation at 400× *g* for 5 min at 4 °C and resuspended in 400 µL Assay buffer BA. Then 100 µL of cell suspensions were incubated with Mitopotential Working Reagent for 20 min in the incubator 37 °C with 5% CO_2_, following incubation they were added 5 ul of the 7-AAD dye and put at room temperature for 5 min in the dark. Apoptosis was determined by membrane potential that was measured by Muse Analysis machine (Guava Muse Cellular Analyser, LO86, Luminex, Austin, TX, USA).

### 2.7. Statistical Analysis

Data are expressed as mean ± SD, unless stated otherwise, and each point shown on the graph represents an independent experiment. Data were recorded on a data collection form and entered on a Microsoft Office Excel^®^ (2013) spreadsheet. Statistical tests were performed using GraphPad Prism version 6.00 software (GraphPad Software, San Diego, CA, USA). A chi-squared test was used to test for significance, and a priori level of *p* < 0.05 was considered statistically significant.

## 3. Results

### 3.1. Antiproliferative Activity of Metformin under Different Glucose Concentrations

The effect of different glucose environments on the antiproliferative activity of metformin was measured using the MTT assay (TOX1-1KT, Sigma-Aldrich, St. Louis, MO, USA). MCF-7 cells were grown in NG, EG, and HG media and treated with a serial dilution of metformin for 24 h and 48 h. Non-treated (vehicle control) samples showed a stable proliferation rate, confirming that the glucose concentrations used had no effect on cellular growth. However, metformin treatment appeared to inhibit cellular proliferation in a concentration and time-dependent manner (Figure 1A,B), with a concentration of 85 µM showing significant inhibition. At 24 h, MCF-7 cells treated with metformin in EG media showed, albeit not significantly different, the highest percentage of cell death compared to NG and HG media, with NG producing the lowest percentage of cell death. However, at 48 h, MCF-7 cells in an NG environment produced a significantly higher percentage of cell death compared to those exposed to an EG and HG environment (*p* = 0.0039), with HG producing the lowest percentage.

### 3.2. The Effect of Glucose Concentrations and the Level of Oxidative Stress

Lack of glucose and nutrient starvation may lead to the production of oxidative stress and thus produce cell death. Thus, it is important to ensure that observed cell death was, in fact, due to metformin treatment and not induced by starvation-induced oxidative stress. Therefore, the oxidative stress level in NG media was measured using the reactive oxygen metabolites test (d-ROMs), which measures the total amount of hydroperoxides [19]. Treatment with doxorubicin, positive control produced a significantly higher level of oxidative stress compared to NG and EG. Interestingly, there was no difference in the oxidative stress level between NG and EG media (Figure 2), confirming that the observed cell death was, in fact, due to metformin treatment.

### 3.3. Effect of Glucose on the Expression Level of Metformin Target Protein

mTOR and BCL-2 proteins were reported to be potential targets for the antiproliferative activity of metformin [20]. Thus, the effect of different glucose levels and metformin treatment on the expression level of these proteins was measured by Western blot analysis (Figure 3). The color intensity of the protein band from the Western blot represents the protein expression level, which was quantified using Image Analyzer Software (ChemiDoc MP Imaging System by Bio-Rad, ImageLab6.1.). Protein expression was determined by calculating the ratio of mTOR and BCL-2 to the housekeeping protein alpha-tubulin.

While the level of mTOR and BCL2 protein expressions of metformin-treated cells in elevated glucose media were significantly reduced compared to control (*p* = 0.04, and *p* = 0.02, respectively) (Figure 3A), cells from low normal glucose media reduced mTOR (*p* = 0.01), but not BCL2 (*p* = 0.8) (Figure 3B). However, there was no difference in the mTOR (*p* = 0.15) and BCL2 (*p* = 0.27) protein expression levels of metformin-treated cells from HG media (Figure 3C).

### 3.4. Metformin-Induced Autophagy under Different Glucose Environments

The ability of metformin and glucose to induce autophagy was determined using the autophagosome detection method, which quantifies the number of autophagocytosed cells. Metformin was able to induce autophagy in EG media that was significantly increased in HG, and expectedly, NG media alone induced a significantly high level of autophagy in MCF7. However, treatment with metformin significantly reduced the measured autophagy level (Figure 4). While NG alone was shown to induce autophagy in MCF7 cells, the explanation as to how metformin was able to inhibit autophagy in cells grown in NG remains unclear.

### 3.5. Metformin-Induced Apoptosis

Cellular apoptosis was measured using the fluorescence Caspase 3/7 assay that was analyzed by Muse Cell Analyzer. MCF-7 cells in an NG media showed a significant level of apoptosis that was reduced following a 24 h treatment of metformin (*p* = 0.04) (Figure 5A). In HG media, 24 h treatment with metformin-induced a significant level of cellular apoptosis compared to the non-treated control (*p* = 0.04) (Figure 5A). Interestingly, the level of metformin-induced apoptosis in HG is significantly less than in NG media alone. However, a 48 h metformin treatment in both NG and HG induced significant levels of cellular apoptosis (*p* = 0.012 and *p* = 0.043, respectively) (Figure 5B).

### 3.6. Mitochondrial Membrane Potential following Metformin Treatment

A change in mitochondrial potential is thought to be an early hallmark of apoptosis. The commercially available Mitopotential Assay kit (Muse® Mitopotential Kit, MCH100110, 9 Luminex Corporation, Austin, TX7 8727 USA) was used to determine the depolarization of mitochondrial potential as an indication of cellular apoptosis. Interestingly changing the glucose concentration of the media for 24 h to either HG or NG significantly increased the depolarization of mitochondrial membrane potential to approximately half-fold and one-fold, respectively (Figure 6A). Furthermore, 24 h metformin treatment in both HG and NG media significantly increased the depolarization. However, following the 48 h incubation, the mitochondrial membrane depolarization potential was decreased in both NG and HG media, but mitochondrial membrane depolarization was significantly increased to approximately 4-fold when metformin was included in the NG (*p* = 0.001) and HG media (*p* = 0.001) (Figure 6B).

## 4. Discussion

The current study investigated the effect of glucose on the antiproliferative activity of metformin against MCF-7. The results showed that at 24 h treatment, metformin, in a concentration-response manner, was able to significantly reduce cellular proliferation of MCF-7 cells in both EG and HG environments compared to the NG environment. However, at 48 h treatment with metformin, cells grown in the NG media showed the highest antiproliferative activity compared to the EG and HG environments. In the low normal (NG) environment, basal levels of ROS may halt or reduce cellular proliferation [21]. However, NG alone did not reduce cellular proliferation, which supports the theory that the observed effect was likely due to metformin. The levels of ROS production from both NG and HG media were tested and showed no significant difference, which further supported the idea that the observed antiproliferative activity was a result of metformin treatment.

mTOR, a protein kinase that regulates protein synthesis and cell growth in response to growth factors, nutrients, energy levels, and stress, is a putative target for metformin [22]. In comparison to control, Western blot analysis of cells treated with metformin in NG and EG, but not HG, showed reduced mTOR protein expression, indicating that metformin under normal (NG) and elevated glucose (EG) environment targets mTOR in its anticancer mechanism. However, there is no consensus on whether metformin targets mTOR or not as an explanation for its putative antiproliferative activity. For example, Deng et al., claimed that metformin achieves its antiproliferative activity independently of mTOR [6], whereas Zakikhani et al., suggested that the antiproliferative activity of metformin depends on the tumor-suppressor liver kinase B1 (LKB1) pathway [23]. Therefore, the current results suggest that the antiproliferative mechanism of metformin depends on the glucose environment. Hence, it can be assumed that in an HG environment, metformin achieves its antiproliferative activity via an mTOR-independent mechanism such as STAT3 phosphorylation, and it is a potential target that is widely supported as the basis for the antiproliferative effects of metformin [23,24].

Furthermore, the protein expression BCL-2, a family of proteins that regulate apoptosis [25], is decreased in NG but not in HG or LG. Interestingly, in NG, metformin reduces both mTOR and BCL-2 protein expressions, whereas in LG, it only reduces mTOR but not BCL-2, and in HG, it does not change the expression of either protein. Therefore, the change in the glucose environment might not impact the potency of the antiproliferative activity of metformin, but rather its mechanism of action, which further confirms that the antiproliferative activity of metformin is strongly dependent on the glucose environment.

Interestingly, NG alone induced significant levels of autophagy and apoptosis. The latter increased with metformin treatment and reached an approximate level of four times the cells maintained in EG by 48 h. A number of studies have shown that maintaining cells in a glucose-starved, or NG, environment resulted in the phosphorylation of p53 and inhibition of mTOR, the latter also confirmed in this study, and that autophagy is enhanced via translocation of p53 to the nucleus where it induces the expression of several autophagy genes [26,27,28]. However, the present results show that metformin treatment reduced autophagy in NG but induced it in EG and HG, possibly through a p53-related mechanism. Yi et al. have reported that metformin interacts with cancer cells in both p53-dependent and p53-independent ways [29].

In the current study, NG alone induced cellular apoptosis in MCF-7 cells, and treatment with metformin further increased apoptosis in these cells. While LG-induced apoptosis is possibly achieved via activating pro-apoptotic genes [30], metformin, as supported by the data on the effects on mitochondria membrane potential, induces apoptosis via a mitochondria-associated pathway. Mitochondria contain key regulators of cell death processes, including via apoptosis [31], and changes in mitochondria function may be used as indicators of cell death and stress [32]. Loss of mitochondrial membrane potential is often, but not always, associated with early stages of apoptosis [32]. Thus, depolarization of inner mitochondrial membrane potential is a reliable indicator of mitochondrial dysfunction and cellular health [33,34]. Therefore, the results indicate that metformin induced apoptosis in MCF-7 cells via a mitochondria-dependent pathway. The current results warrant further study to determine the exact antiproliferative mechanism of metformin and how they are impacted by different glucose concentrations.

## 5. Conclusions

In conclusion, the present study investigated the effect of glucose on the antiproliferative activity of metformin over different time points using human breast cancer cells (MCF-7). The results confirm the antiproliferative potential of metformin in three different glucose environments. This is the first study to show that the antiproliferative mechanism of metformin is glucose-dependent and that the antiproliferative activity of metformin in NG is achieved through mTOR, but in HG, the antiproliferative activity is independent of mTOR.

## 6. Study limitations

This is a cell-culture (in vitro) study, and hence, the findings may not be directly applicable in clinical practice. However, this is a proof of concept, and the results in this study will form the basis for animal and clinical testing. Another drawback is that the results were based on one breast cancer cell line (MCF-7) only and that comparative studies with other cell lines should also be pursued. On the other hand, the current study investigated the effects of a range of glucose concentrations from 5.5 mM to 25 mM and also a concentration of metformin, 85 μM, and, although comparatively high versus those expected in the plasma when used as an anti-hyperglycemic drug, likely reflects the intracellular concentration in cells, which are albeit dependent on the expression levels of organic cation transporters [2].

## Figures and Tables

**Figure 1 pharmaceutics-15-02186-f001:**
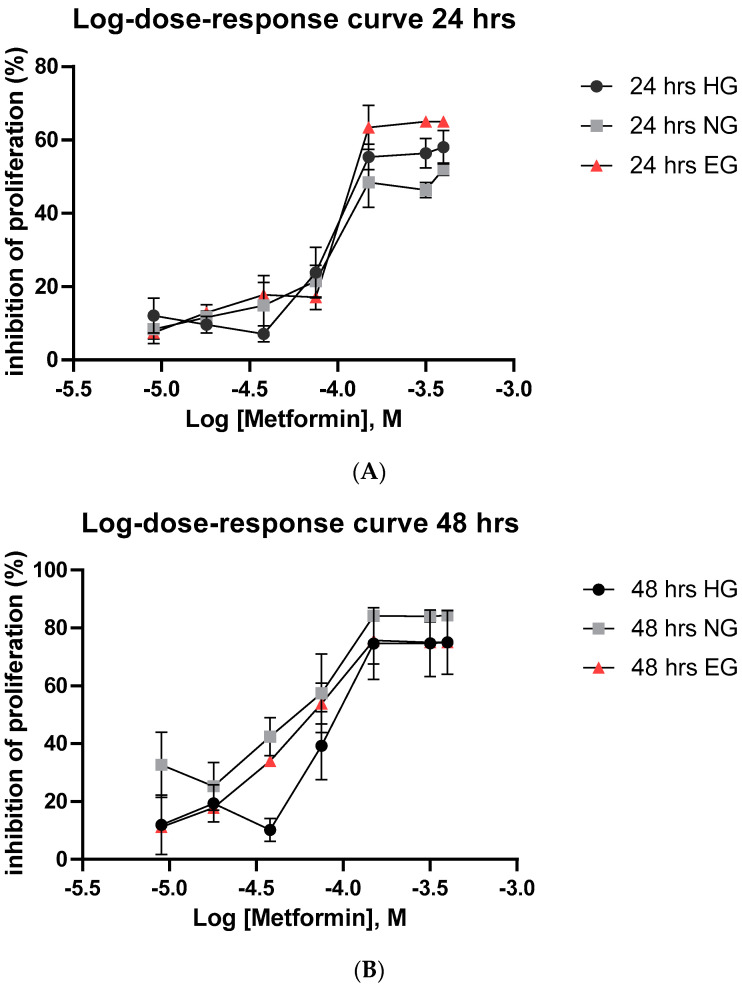
The effect of glucose environment on the antiproliferative activity of metformin in MCF-7 breast cancer cells. The graph shows the effect of different glucose environments on the antiproliferative activity of metformin expressed as percentage of cell death vs drug concentration. Cells were grown in low normal (NG), elevated (EG) or High Glucose (HG), and then treated with serial dilutions of metformin (10 μM to 150 μM) for 24 h (**A**), and 48 h (**B**). There was no significant difference between treatments at 24 h (**A**), but at 48 h (**B**), NG environment showed a significantly (*p* = 0.0039) higher cell death compared to EG and HG. Data are expressed as mean ± SD (N = 6–8 for each presented value).

**Figure 2 pharmaceutics-15-02186-f002:**
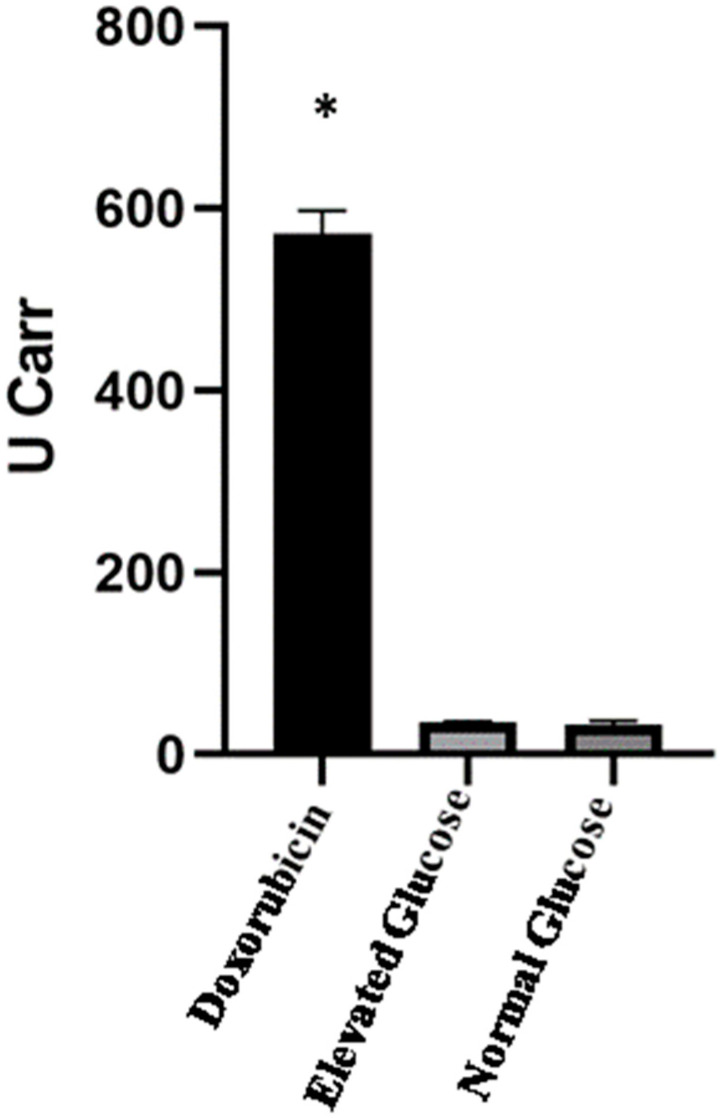
Glucose concentrations and oxidative stress. The graph shows the total amount of hydroperoxides as measured by the reactive oxygen metabolites test (d-ROMs). The results show a significantly high level of hydroperoxides in the doxorubicin treated samples (*p* < 0.001), but no difference in the oxidative stress levels between EG and NG (*p* = 0.4). Data are expressed as mean ± SEM; N = 6. * Represent a significant difference.

**Figure 3 pharmaceutics-15-02186-f003:**
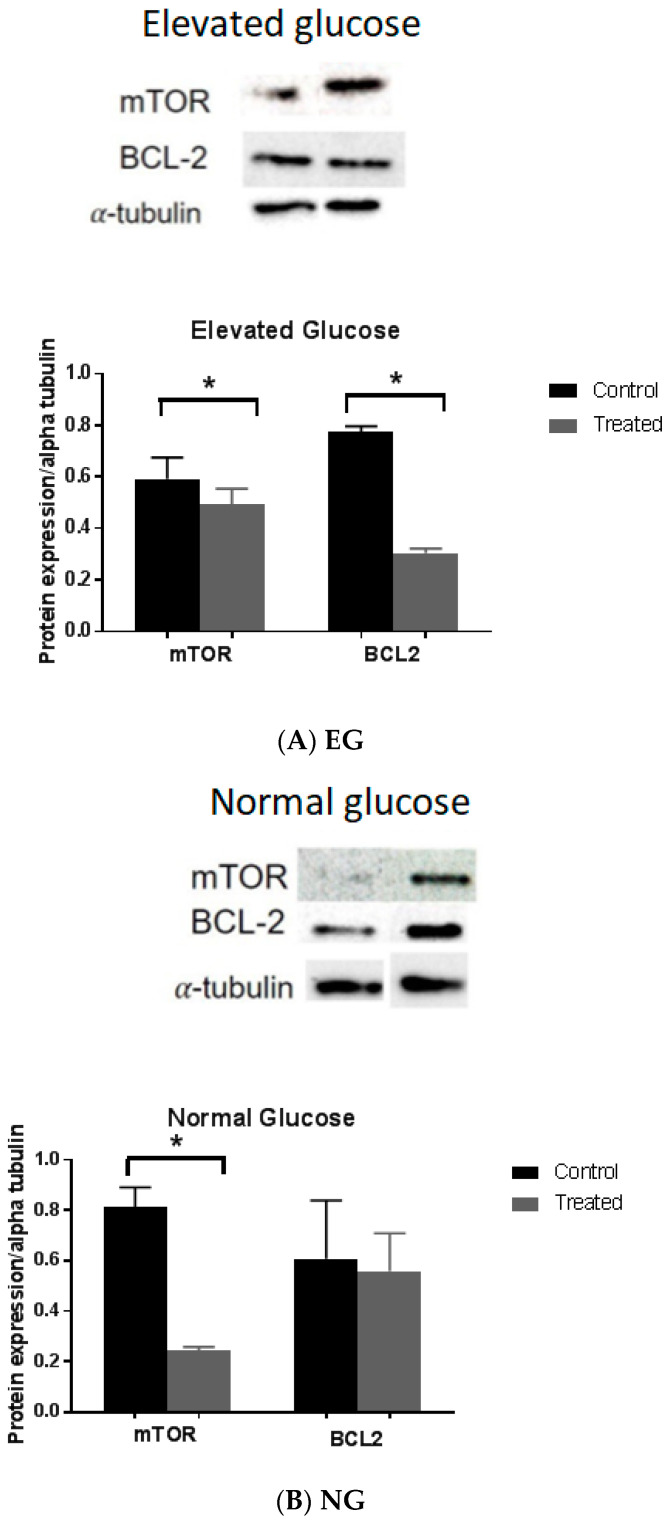
Effect of glucose level on protein expression. MCF-7 cells were grown in different glucose levels; Elevated Glucose (**A**), Low Normal Glucose (**B**), and High Glucose (**C**). Cells were then treated with 85 µM of metformin for 24 h. The western blot images above show mTOR, BCL2 and alpha-tubulin protein expressions from each of the 3 different glucose environments. The graphs represent relative expression intensity of mTOR and BCL-2 in relation to alpha-tubulin from control MCF-7 cells (non-treated) and 24 h treatment of cells with 85 µM metformin. Compared to control, metformin treatment in elevated glucose significantly reduced protein expression levels of mTOR (*p* = 0.04) and BCL2 expression (*p* = 0.02). However, metformin treatment in low normal glucose reduced mTOR (*p* = 0.01), but not BCL2 (*p* = 0.8), and no difference observed in high glucose. Data are expressed as mean ± SD; N = 3. * Represent a significant difference. The uncropped bolts are shown in Appendix A.

**Figure 4 pharmaceutics-15-02186-f004:**
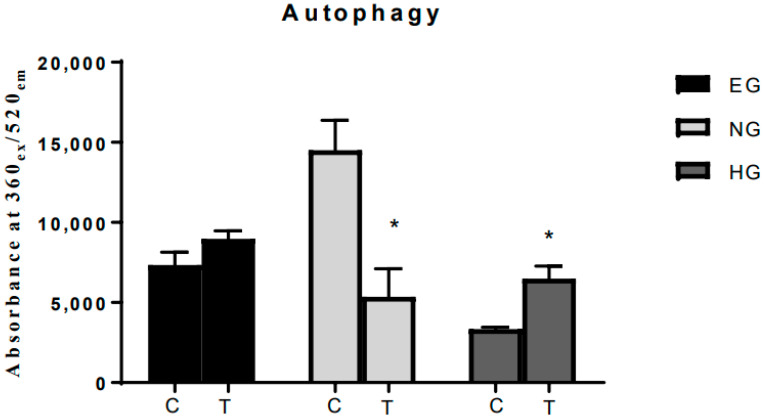
Effects of metformin and glucose on autophagy. The graph above shows the autophagy induction ability of metformin and different glucose environment on MCF-7 as measured by fluorescence absorbance that reflects autophagocytosed cells. The letter C represents control for cells untreated with metformin and the letter T designates cells treated with 85 µM of metformin for 24 h. A normal low glucose environment alone induced autophagy in MCF-7 (*p* = 0.001) that was reduced with metformin treatment. In high glucose environment, treatment with 85 µM of metformin significantly induced autophagy (*p* = 0.04). Data are expressed as mean ± SD; N = 6. * Represent a significant difference.

**Figure 5 pharmaceutics-15-02186-f005:**
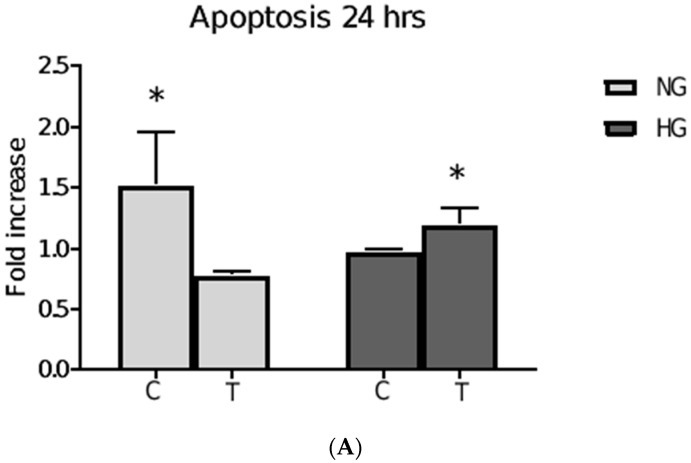
Metformin effect on apoptosis. The graph above shows the apoptotic activity of metformin under different glucose environment over different time points. Cells incubated in NG and HG media and treated with 85 µM of metformin for 24 h (**A**) and for 48 h (**B**).The apoptotic activity in the graph shows a fold increase relative to control cells from EG media. The letter C is control for untreated cells that were growing in their perspective glucose level and the letter T (treated) designates cells treated with 85 µM metformin. At 24 h metformin treatment significantly reduced apoptosis in NG (0.04), but increased it in HG (*p* = 0.04) (**A**). (**B**) shows the increased levels of apoptosis following 48 h treatment with metformin NG (*p* = 0.012) and HG (*p* = 0.043).Data are expressed as mean ± SD; N = 3. * Represent a significant difference.

**Figure 6 pharmaceutics-15-02186-f006:**
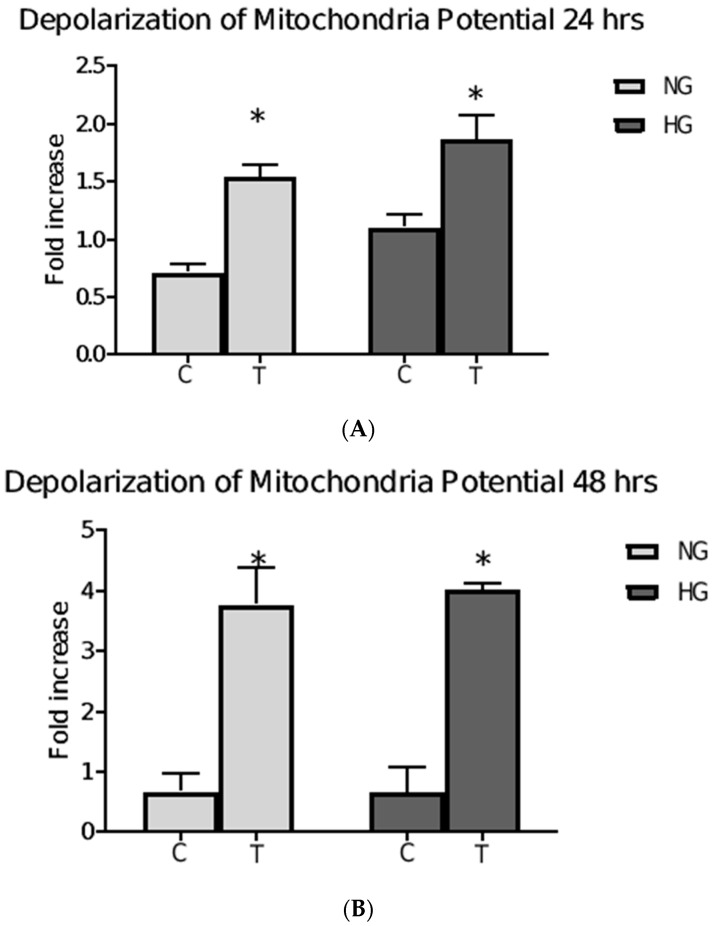
Effect of metformin on mitochondrial membrane potential. The graphs show the fold increase (relative to EG) of depolarization of mitochondrial membrane potential following treatment with 85 µM metformin (T) and control (C) for 24 h (**A**) and 48 h (**B**) in cells from NG and HG. Metformin treatment for 24 h in both HG and NG significantly increased the depolarization (*p* = 0.004, and *p* = 0.003, respectively) (**A**). The depolarization was further increased up to four folds in NG (*p* = 0.001), and in HG (*p* = 0.001) compared to control (C) (**B**). Data are expressed as mean ± SD; N = 3. * Represent a significant difference.

## Data Availability

The data that support the findings of this study are available from the corresponding author upon reasonable request. Some data may not be made available because of privacy or ethical restrictions.

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
