# Peer review of "The Effect of Different Glucose Concentrations on the Antiproliferative Activity of Metformin in MCF-7 Breast Cancer Cells"

_pharmaceutics, 2023, doi:10.3390/pharmaceutics15092186_

Round 1
Reviewer 1 Report (Previous Reviewer 1)
The revised version improved substantially. No further comments.
Author Response
Thank you
Reviewer 2 Report (New Reviewer)
Although the study is interesting and generally well written, it presents several flaws that must be resolved. See:
Lines 35-49: It deserves to be pointed out that metformin can also increase cancer cell sensitivity to chemotherapeutics in a glucose-independent manner (as recently reviewed PMID: 36361682) futher highlighting its anti-cancer effects.
2.1. Ethical statement: this chapter is superfluous
Line 89: which cells?
2.3. Oxidative stress assay: product code of Plasma Antioxidant Test kit must be reported
2.4. Western blotting: product code and dilution of primary antibodies must be reported
Figure 3: the quality of western blot is very low and it is not written what is loaded in each lane. Representative western blot must be shown
Round 2
Reviewer 2 Report (New Reviewer)
the manuscript has been significantly improved and can be accepted in the present form
This manuscript is a resubmission of an earlier submission. The following is a list of the peer review reports and author responses from that submission.
Round 1
Reviewer 1 Report
The effect of glucose on the antiproliferative activity of metformin
Sholpan Nurzhan, Zhibek Bekezhankyzy, Hong Ding, Nurken Berdigaliyev, Shynggys Sergazy, Alexandr Gulyayev, Zarina Shulgau, Christopher R. Triggle, Mohamad Aljofan
General comment
The authors studied the antiproliferative and apoptotic effects of metformin in relation to normal, elevated and highly elevated glucose concentrations in MCF-7 breast cancer cells in vitro. This study is of interest but contains several uncertainties that should be clarified prior to publication.
Special comments
Title
The reviewer suggests to improve the title, which is not precise:
Better: The effect of different glucose concentrations on the antiproliferative activity of metformin in MCF-7 breast cancer cells in vitro
Introduction
Line 48
The authors mentation that IGF1 and AMPK would function as negative regulators of mTOR pathway. IGF1 via activation of the kinase AKT enhances the activity of mTORC1. mTORC2 also activates AKT-mediated activation of mTORC1.
Furthermore, the potential impact of metformin on the activity of mTORC1 has been published some years ago.
Melnik BC, Schmitz G (2014) Metformin: an Inhibitor of mTORC1 Signaling. J Endocrinol Diabetes Obes 2(2): 1029.
Methods
1) A comparison between normal and malignant mammary epithelial cells is missing
The authors should provide data of a non-malignant mammary epithelial cell to allow a comparison of metformin effects between normal and malignant epithelial cells of the mammary gland.
Is there a difference between normal cells and malignant breast cancer cells in metformin uptake and biological responses (proliferation/apoptosis)?
This may be important because a breast cancer will not affect the whole mammary tissue.
2) Missing approximation of metformin doses that reach the mammary gland
Do the authors have data on metformin tissue concentrations after oral uptake of metformin in common clinical dosages of 2000 mg/day?
According to the literature, the mean plasma concentrations of metformin fluctuate between 0.4 and 1.3 μg/mL with the daily dosage of 2000 mg, and concentrations above 5 μg/mL are considered to be elevated.
Graham GG, Punt J, Arora M, Day RO, Doogue MP, Duong JK, Furlong TJ, Greenfield JR, Greenup LC, Kirkpatrick CM, Ray JE, Timmins P, Williams KM. Clinical pharmacokinetics of metformin. Clin Pharmacokinet. 2011 Feb;50(2):81-98. doi: 10.2165/11534750-000000000-00000. PMID: 21241070.
Duong JK, Kumar SS, Kirkpatrick CM, Greenup LC, Arora M, Lee TC, Timmins P, Graham GG, Furlong TJ, Greenfield JR, Williams KM, Day RO. Population pharmacokinetics of metformin in healthy subjects and patients with type 2 diabetes mellitus: simulation of doses according to renal function. Clin Pharmacokinet. 2013 May;52(5):373-84. doi: 10.1007/s40262-013-0046-9. PMID: 23475568.
Li L, Guan Z, Li R, Zhao W, Hao G, Yan Y, Xu Y, Liao L, Wang H, Gao L, Wu K, Gao Y, Li Y. Population pharmacokinetics and dosing optimization of metformin in Chinese patients with type 2 diabetes mellitus. Medicine (Baltimore). 2020 Nov 13;99(46):e23212. doi: 10.1097/MD.0000000000023212. PMID: 33181704; PMCID: PMC7668473.
Wilcock and Bailey studied the accumulation of metformin by tissues of the normal and diabetic mouse. 1h after administration of 50 mg metformin/kg body weight they detected 154 M/L in the liver, 137
M/L in the submaxiallary salivary gland but only 15
M/L in white fat, respectively. Unfortunately, no data were available for mammary glands.
Wilcock C, Bailey CJ. Accumulation of metformin by tissues of the normal and diabetic mouse. Xenobiotica. 1994 Jan;24(1):49-57. doi: 10.3109/00498259409043220. PMID: 8165821.
For an approximation of metformin concentrations for in vitro exposure more data should support the experimental setting.
3) mTOR measurements do not allow conclusions on TORC1- and mTORC2-dependent metabolic and proliferative effects
The authors only determined mTOR, the core kinase of the mTORC1 and mTORC2 complex.
mTORC1 is of key importance for the regulation of cell anabolism, cell growth and apoptosis.
mTORC1 activity can be extrapolated by measuring the phosphorylation of S6 kinase (S6K1-P), the downstream target of mTORC1.
Discussion
The reviewer wonders why the authors missed the highly related paper:
Shehata M, Kim H, Vellanki R, Waterhouse PD, Mahendralingam M, Casey AE, Koritzinsky M, Khokha R. Identifying the murine mammary cell target of metformin exposure. Commun Biol. 2019 May 20;2:192. doi: 10.1038/s42003-019-0439-x. PMID: 31123716; PMCID: PMC6527562.
Furthermore, the initial enthusiasm of metformin´s anti-breast cancer activity was tempered after disappointing results in randomized controlled trials, particularly in the metastatic setting.
Cejuela M, Martin-Castillo B, Menendez JA, Pernas S. Metformin and Breast Cancer: Where Are We Now? Int J Mol Sci. 2022 Feb 28;23(5):2705. doi: 10.3390/ijms23052705. PMID: 35269852; PMCID: PMC8910543.
Nevertheless, metformin may have breast cancer-protective effects.
Naseri A, Sanaie S, Hamzehzadeh S, Seyedi-Sahebari S, Hosseini MS, Gholipour-Khalili E, Rezazadeh-Gavgani E, Majidazar R, Seraji P, Daneshvar S, Rezazadeh-Gavgani E. Metformin: new applications for an old drug. J Basic Clin Physiol Pharmacol. 2022 Dec 7;34(2):151-160. doi: 10.1515/jbcpp-2022-0252. PMID: 36474458.
Intriguingly, metformin via reducing the oncogenic long non-coding RNA H19 may exert benefical epigenetic effects that should also be mentioned in the discussion.
Zhong T, Men Y, Lu L, Geng T, Zhou J, Mitsuhashi A, Shozu M, Maihle NJ, Carmichael GG, Taylor HS, Huang Y. Metformin alters DNA methylation genome-wide via the H19/SAHH axis. Oncogene. 2017 Apr 27;36(17):2345-2354. doi: 10.1038/onc.2016.391. Epub 2016 Oct 24. PMID: 27775072; PMCID: PMC5415944.
Chen J, Qin C, Zhou Y, Chen Y, Mao M, Yang J. Metformin may induce ferroptosis by inhibiting autophagy via lncRNA H19 in breast cancer. FEBS Open Bio. 2022 Jan;12(1):146-153. doi: 10.1002/2211-5463.13314. Epub 2021 Oct 27. PMID: 34644456; PMCID: PMC8727937.
Gholami M, Klashami ZN, Ebrahimi P, Mahboobipour AA, Farid AS, Vahidi A, Zoughi M, Asadi M, Amoli MM. Metformin and long non-coding RNAs in breast cancer. J Transl Med. 2023 Feb 27;21(1):155. doi: 10.1186/s12967-023-03909-x. PMID: 36849958; PMCID: PMC9969691.
Minor comments
Please unify SI units, such as uM versus M throughout the text.
Some spelling errors and variation in units such as uM versus M
Reviewer 2 Report
This is an interesting in vitro study, which lacks in the applicability to clinical practice.
Major points:
1. please explain in details how the glucose levels applied in the cell cultures (from 5.5 mM to 25 mM) are representing the glucose levels in human subjects.
2. please convert the normal glucose levels applied in the cell cultures into the glucose levels for human subjects.
3. please convert the elevated glucose levels applied in the cell cultures into the glucose levels for human subjects.
4. please convert the high glucose levels applied in the cell cultures into the glucose levels for human subjects.
5. please explain in details how the concentration of metformin (85 M) applied in the cell cultures are representing the concentration of metformin given in clinical practice to patients with type-2 diabetes.
none